# Poverty, Household Structure and Consumption of Foods Away from Home in Peru in 2019: A Cross-Sectional Study

**DOI:** 10.3390/foods11172547

**Published:** 2022-08-23

**Authors:** Michelle Lozada-Urbano, Franklin Huamán, Yanira Xirinachs, Oriana Rivera-Lozada, Aldo Alvarez-Risco, Jaime A. Yáñez

**Affiliations:** 1South American Center for Education and Research in Public Health, Universidad Norbert Wiener, Lima 15046, Peru; 2Facultad de Ingeniería Económica, Estadística y Ciencias Sociales, Universidad Nacional de Ingeniería, Lima 15333, Peru; 3Escuela de Economía, Universidad de Costa Rica-UCR, Ciudad Universitaria Rodrigo Facio Brenes, San José 02060, Costa Rica; 4Carrera de Negocios Internacionales, Facultad de Ciencias Administrativas y Económicas, Universidad de Lima, Lima 15023, Peru; 5Facultad de Educación, Carrera de Educación y Gestión del Aprendizaje, Universidad Peruana de Ciencias Aplicadas, Lima 15023, Peru; 6Gerencia Corporativa de Asuntos Científicos y Regulatorios, Teoma Global, Lima 15073, Peru

**Keywords:** food consumed, away from home, families, spending, logit, ENAHO, Peru

## Abstract

The aim of this study was to evaluate the probability of buying food away from home according to the type of household using the logit model, as well as the sociodemographic characteristics of the heads of household, and how much income expenditure represents. A cross-sectional study was carried out using the National Household Survey (ENAHO) 2019 database. After joining the database, the household type variables were created. To calculate the probability with the “logit” model of purchase, the variables—family size, income, types of household, and total expenditure—were selected as a measure of the purchasing power of the family. A statistically significant association (*p* < 0.05) was found between the probability of consumption and the variables: age of household members, predominance, nuclear without children–married, nuclear with children–cohabitant, nuclear with children–widowed, nuclear with children–separated, extended, compounded, poor not extreme, and not poor. The bulk of families was represented by nuclear families (61.97%). The highest expenditure in the CFAH was for families defined as composite with a yearly average of USD 1652.89 (equivalent to PEN 5520.67). Observing the expenditure on food consumed outside the home through the composition of households can allow a better approach to offer educational measures. This information can be helpful to developers of educational issues.

## 1. Introduction

Food choice is influenced by consumption patterns [1,2,3,4] and eating and cooking behaviors, which have changed dramatically in 20 years, partially influenced by rapid urbanization [5] and increased awareness of nutritional labeling related to the presence of transgenics [2]. Furthermore, consumers seek cleaner labels and the presence of bioactive compounds because of their preventive claims [6,7,8]. There is an increased awareness of the negative effect of pesticides [3] and the predation risk caused by the gastronomic boom of countries such as Peru that promote culinary tourism [4]. Sustainability and sustainable development goals play a critical role in the constant supply chain of food [9,10], recycling of food waste [11], waste reduction [12,13], and reduced carbon footprint [12,14,15] and water footprint [12,13,15]. All these factors, together with the exponential growth of restaurants, provide more options for families to engage in greater consumption of food away from home (CFAH) [16,17]. Meals bought and eaten outside the home have been shown to have poor nutritional quality [18,19,20], which has been linked with an association between eating in restaurants and higher incidence of overweight in children [21] and higher risk of overweight and obesity in adults [22,23]. In terms of gender, Chinese males were more likely to increase their body mass index (BMI) when eating breakfast or dinner away from home [17], with a higher incidence of other chronic diseases [18].

Because the nutritional composition of homemade meals and the ones prepared in restaurants differ in the concentrations of total fat, protein and the sodium to potassium ratio [20], it has been reported that people who engage in CFAH have a lower intake of protein, iron, and dietary fiber [24] but at the same time have a higher intake of calories, fat, and cholesterol [25,26]. This causes a higher intake of foods rich in sugar and fat [5] that makes it more difficult for consumers to comply with regimens to control their body weight and improve their blood parameters [27]. Unhealthy or low-quality diets cause micronutrient deficiencies and increase overweight and obesity, allowing the appearance of non-communicable diseases such as coronary heart disease, stroke, and diabetes [28]. Conversely, it has been reported that when a person decreases CFAH, the quality of the diet improves and there is a reduction in BMI and total body fat [29]. A study in Argentina reported that in order to better plan national nutritional objectives, it was relevant to measure food consumption outside the home according to age groups, gender, and types of households [1]. There is limited information regarding CFAH in Peru and the correlation between socioeconomic factors and type and composition of households. The aim of this study was to evaluate the probability of buying food away from home according to the type of household using the logit model, as well as the sociodemographic characteristics of the heads of household, and how much income expenditure represents using data from the National Household Survey (ENAHO) 2019 database. This would allow us to predict consumption patterns to aid in economic predictions, as well as potential correlation with the progression of obesity and non-communicable diseases.

The structure of the present article is as follows: the literature review is discussed in Section 2. The approach used to gather and evaluate the data is described in Section 3. At the same time, the results and findings are presented in Section 4. Section 5 deals with the discussion and Section 6 with the Conclusions.

## 2. Literature Review

A higher income provides a higher purchasing power, which has been reported to increase the consumption of processed foods with high amounts of saturated fats [30,31]. Food choice is a responsibility [32], and it has been reported that with increasing age and socioeconomic level there is less time to prepare foods at home [16], and an increase in food consumed away from home has been reported [31,33]. Furthermore, when there is a higher number of working adults in the household, there is a reduction in the time available for preparing meals, which increased CFAH because of the quick access to food [1,34,35]. Another important aspect is the trust of the public in the food safety standards of restaurants and premade meals. For instance, in the United States more people tend to buy premade meals because of the high food safety standards [35]. On the contrary, a study in Bangladesh reported that food adulteration is a major concern in the population, and adults tend to spend less money consuming foods away from home and spend more time preparing meals at home [36]. It has been reported by the United States Department of Agriculture that the countries that have promoted important changes, such as an increase in income, have also allowed the purchasing power of food to increase [37]. Furthermore, countries respond differently to changes in food prices according to their income, with higher sensitivity in countries with lower income [37]. The General Theory of Employment, Interest, and Money states that patterns in consumption are directly linked with changes in consumption, including that of food [38].

Another important aspect to consider is the generation of food waste. It has been reported that the delivery of food to the home during the COVID-19 pandemic may have improved purchasing decisions and reduced the generation of food waste [39]. In the Mexican context, the COVID-19 pandemic changed families’ food management, since it increased the purchase of groceries, causing a reduction in food waste [40]. Food waste became evident in consumers who stayed at home for 24 h during the COVID-19 quarantine, which represents a major barrier to achieving food security and reducing hunger [39]. Studies that have assessed the point of sale where food is purchased found that the price in restaurants affected the main meal selection, with a significant correlation based on spending capacity [41]. However, when the point of sale was fast food establishments, no change was noticeable [41]. Buying food online satisfied the group of people who are familiar with this type of purchase, with those who show a higher educational level, and with those who think that this channel is easy to use [42].

Because the nutritional composition of homemade meals and the ones prepared in restaurants differ in the concentrations of total fat, protein and the sodium to potassium ratio [20], it has been reported that people who engage in consumption of food away from home (CFAH) have a lower intake of protein, iron and dietary fiber [24], but at the same time have a higher intake of calories, fat, and cholesterol [25,26]. This causes a higher intake of foods rich in sugar, fat, and sodium [5] and makes it more difficult for consumers to comply with regimens to control their body weight and improve their blood parameters [27]. Conversely, it has been reported that when a person decreases CFAH, the quality of the diet improves and there is a reduction in BMI and total body fat [29]. A study in Argentina reported that in order to better plan national nutritional objectives, it was relevant to measure food consumption outside the home according to age groups, gender, and types of households [1]. There is limited information regarding CFAH in Peru and the correlation between socioeconomic factors and type and composition of households.

## 3. Methodology

### 3.1. Database and Construction of Variables

A cross-sectional study was performed using secondary source analysis from the National Household Survey (ENAHO) 2019 database, compiled by the Instituto Nacional de Estadística e Informática (INEI) [43]. The survey was carried out at the national level, in urban and rural areas, in the 24 departments of the Peru and in the Constitutional Province of Callao. The sampling was probabilistic of areas, stratified, multi-stage, and independent in each department. The survey is freely accessible and is divided into the following modules: health, education, and the summary variable, and it was developed based on the economic model of the demand for food and nutrition [44]. The following calculated variable was used: food consumed away from home obtained from restaurants, street vendors, and others (7 questions).

For the measurement of households engaging in CFAH, we used the variable “expenses for food consumed outside the home-g05hd” of the summary module. This variable allowed us to know if a household, with its various characteristics, had consumed food outside the home during 2019. In the first stage of our study, we focused on analyzing the characteristics of households that consume food outside the home in comparison to those that did not. In that sense, the following questions related to our variables of interest were used: (a) households with people under 14 years old; (b) households with people over 60 years old; (c) households with a greater presence of men; (d) households with a greater presence of women; (e) type of household (single person, nuclear, extended, compound, other); (f) household with male/female head of household; (g) household with head of household according to marital status (cohabitant, married, widowed, divorced, separated). The types of household are defined as:Single person: one person (male or female) living alone.Nuclear: household with a conjugal nucleus.Extended: presents a complete or incomplete conjugal nucleus, plus other relatives of the head of household.Compound: presents a complete or incomplete conjugal nucleus, may or may not have other relatives of the head of household, and may not be relatives of the head of household.Non-nuclear: households made up of two or more people without a conjugal nucleus with the presence of other relatives and/or non-relatives of the head of household.

This type of household is made up of two or more people. It has been previously reported that the number of household members, genre, and age group are relevant at the time of deciding food purchase and expenditure [45].

Then, to construct the indicator (consumption of food away from home or not) of the present study, a “dummy” or dichotomous variable of the variable “expenditure for food consumed away from home-g05hd” was used, where 1 implies that the household does consume food away from home and 0 implies the household does not consume food away from home. It should be noted that this indicator was chosen for easy interpretation and modeling with other variables from the same survey. The variable “expenditure on food consumed away from home” is part of the Summary module (calculated variables). The survey questions related to where the foods were consumed away from home were obtained: from restaurants, street vendors, and others. Finally, to model the probability that a household consumes or does not consume food away from home, the logit model was used with the variables, totaling a *n* = 34,565 of eligible households from the 2019 ENAHO survey.

The main “exposure variable” of quantitative nature, the household that spent on food consumption away from home, was taken for power calculation that was carried out considering a confidence level of 95%. This study corresponds to a secondary source analysis because we used the ENAHO 2019 database. The households that paid for and consumed food outside the home were selected and met the characteristics of the types of households selected (composite family, extended family, nuclear and single person). This database contains a series of variables that are related to the objective of this study.

The characteristics of the households that consumed food outside the home were described in comparison to those that did not. For this, questions related to the variables of interest were used: (a) households with people under 14 years of age; (b) households with people over 60 years of age; (c) households with a greater presence of men; (d) households with a greater presence of women; (e) type of household (single person, nuclear, extended, compound, other); (f) household with male/female head of household; (g) household with head of household according to marital status (living together, married, widowed, divorced, separated).

The inclusion criterion was that the household had paid for and consumed food outside the home and met the characteristics of the types of household studied (nuclear, extended, compound, single-person, or non-nuclear household).

### 3.2. Construction of Indicators

In households with people under 14 years old (extreme_minor), we selected the variables with members under 14 years old (p208a <= 14). In the case of households with people older than 60 years old (extreme_major), the variables with members older than 60 years old were selected (p208a >= 60). For households with a greater presence of men (presence_of_men), the variables with male members over 14 years old were selected (p208a >= 14 & p207 == 1). For households with a greater presence of women (presence_of_women), the variables with female members older than 14 years were selected, (p208a >= 14 & p207 == 2). For type of household (type_household), the variables were selected according to the relationship with the head of household and the number of family members, single-person home (type_household == 1), nuclear household (type_household == 2), extended household (household_type == 3), composite household (household_type == 4), non-nuclear household (household_type == 5). For households with a male/female head of household (p203b), we selected variable p203b. For households with a head of household according to marital status (p207), marital status of the head of household was selected, cohabitant (p207 == 1), married (p207 == 2), widowed (p207 == 3), divorced (p207 == 4), separated (p207 == 5), single (p207 == 6). The ENAHO survey was analyzed according to the expansion factor. The poverty variables were extremely poor, very poor and non-poor. A Logistic regression was applied, and the “Svy” command was used for complex samples, as previously described [46].

### 3.3. Data Analysis

A logit model was used to model the probability that a household consumes or does not consume food away from home, as previously described [46,47]. The type of household, predominance, presence of children (lower end) and adults (higher end) and poverty were used as variables. A total of 34,565 eligible households from the 2019 ENAHO survey were used. The main exposure variable of quantitative nature was the household that spent on food consumption away from home was taken for power calculation that was carried out considering a confidence level of 95%. Then, to construct the indicator for this study, a dummy or dichotomous variable was used for the variable “expenditure for food consumed outside the home-g05hd”, where 1 implied that the household did consume food outside the home, and 0 implied the household did not. It should be noted that this indicator was chosen for its ease of interpretation and modeling with other variables from the same survey. The effect of each variable on the possibility of purchase was obtained. Stata Statistical Software, release 16 (StataCorp LLC, College Station, TX, USA) was used with statistical significance at *p* < 0.05.

## 4. Results

The data analyzed were obtained from the National Household Survey (ENAHO) version 2019 based on household records in Peru. Initially, there were 34,565 households. The participation in consumption was assessed according to the gender of the head of household and the annual expenditure on food consumption outside the home and per capita expenditure. Table 1 shows that when the head of household is a man, he spends more away from home; however, when the per capita spending is compared, women spend more than men. Buying and consuming food outside the home implies spending less time cooking. It has been reported that when the head of household is a woman, there is more time spent preparing meals at home [48,49,50]. In households with a male head, the proportion of households that consume foods away from home is statistically higher compared to households when the head is female. Likewise, the difference in the household spending was statistically significant and no difference was found in the per capita spending.

Single-person and non-core households have negative values; therefore, their probability of consuming food outside the home decreases. When the extended and nuclear households were compared with compounded households, it was found that compounded households have a lower probability of consuming food away from home. The highest average annual expenditure was carried out by compound households, which is consistent with the economies of scale derived for the preparation of meals in households where there is more than one person [51].

The highest average age is for single-person households, followed by extended households with nuclear households showing the lowest age. The highest average expenditure was reported for composite households, followed by extended households, and the lowest average expenditure was reported for single-person households. Single-person households reported the highest per capita spending, followed by non-nuclear households, and the lowest spending was observed for the extended household (Table 2).

Table 3 shows the coefficients and marginal effects of the logit model, with the variable “poverty” divided into extreme poverty, non-extreme poverty, and non-poor, as previously reported [52]. The results of the logit model offer the value of the impact of the variables on the probability of consumption, with positive variables increasing the probability of consumption. As shown in Table 3, the coefficients of the categories single person, separated, widowed and divorced were negative. The nuclear household without children (cohabitant) and nuclear household without children (married) exhibited significant coefficients for the result of the marginal effect. Nuclear households with children (single) had a positive coefficient, as did nuclear households with children (cohabitant), married, widowed, divorced, separated, extended, compound and non-nuclear households, the non-extreme poor, and the non-poor.

The proportion of CFAH spending with respect to the total net income of household is showed in Figure 1. It was observed that according to the poverty level, the highest percentage of CFAH spending was reported in single persons (9%) and the lowest percentage of CFAH spending (4.14%) was reported in the lowest economic stratum: extreme poor.

## 5. Discussion

The aim of this study was to evaluate the probability of buying food away from home according to the type of household using the logit model, as well as the sociodemographic characteristics of the heads of household, and how much income expenditure represents. It has been reported in Argentina and Uruguay that when the head of household is male, there is higher annual expenditure on consumption of food away from home (CFAH) [53,54]. The results of these previous studies correlate with the results of this study in Peru, where it was observed that men have greater preference for CFAH than women (0.813 and 0.766, respectively) a higher statistically significant annual expenditure (PEN 3089.53 for male household heads and PEN 2938.92 for female household heads, *p* < 0.05). The correlation between the gender variable and expenditure has been linked with economic autonomy, which favors the defamiliarization of domestic work as women become economically independent and they spend less time doing housework [53], and to purchasing meals outside the home [54]. Conversely, other studies mentioned that women who work in the non-farm sector are more likely to consume food outside the home [36], and it has been recently reported that stressed females tend to eat more away from home than males in China [55]. The results of this study reported that in households where men predominate and households with only men, there is a statistically higher CFAH spending compared to households with a predominance of women and households with only women.

These results correlate with previous studies, in which men consumed more than women when buying foods outside the home in China [56] and Latin America and the Caribbean [57].

Regarding the type of household, it has been previously reported that extended households present a higher probability of consumption of food away from home [1]. A previous study mentions that the age and purchasing power of the head of household can represent an important point in the consumption of food away from home [58]. This study reported that there are differences in total expenditure, which could be attributed to the age difference, with single-person households having older heads of households, and nuclear households having younger heads of households [58]. The higher per capita expenditure in a single-person household (older head of household) has been reported to be linked with the time constraints for food preparation, with lower expenditure on natural foods and more expenditure on processed products [31]. The food type selection (processed versus natural) has been reported to be correlated with the body mass index and waist circumference of the main buyer [56,59]. Our results report that single-person households spend 9% of their income, extended households 7.53%, and composite households spend 5.79%. The latter would be consistent with the economies of scale derived for the preparation of meals in households where there is more than one person.

There are other factors that influence the head of household’s decision whether to buy food away from home or not [31,60]. One of these factors is working hours, with those working full time spending less on food to prepare at home and more on food consumed outside the home [31,61]. Lima is growing rapidly, and commuting implies hours of travel from one place to another, reducing the time available to prepare meals at home and increasing the choice to consume food outside the home. The value of time also includes factors such as demographic changes, transportation, and bureaucratic procedures, among other things [62]. It has been reported that households that are close to fast food restaurants are more likely to spend an additional 25% of their income on food outside the home [27,63].

The current study reports that non-poor households spend 7.20% of their income, and extreme poor households spend 4.14%. This correlates with previous studies that reported that when the head of household has a higher socioeconomic and educational level, there is higher expenditure on food outside the home [64]. It is important to note that the database contains information about food consumed at home and outside home obtained from charities. However, this information was not part of the scope of this study, but could influence the lower expenditure observed for extreme poor households.

The results of this study will allow researchers and authorities to go beyond the economic data and create better public policies. This type of information is necessary, as this involves the problems caused by CFAH and their effect on health. It has been reported that high expenditure on CFAH in the United States was associated with low dietary quality [65]. Studies in Peru have shown that children’s menus obtained from fast food restaurants had an increase in fat content ranging from 14% to over 60% [66]. Furthermore, it has been reported that over 50% of processed foods in Peru have increased their sugar content in the last 3 years [67]. This is concerning because the incidence of overweight and obesity in children at a very early age and in adults is increasing in Peru, along with problems such as malnutrition and anemia [68,69].

### Limitations and Future Research Directions

One main limitation of the current study is that the data obtained from the database are from 2019, before the COVID-19 pandemic; it is expected that food consumption trends changed during and after the pandemic. Future research should be centered on performing similar studies in other regions or other Latin American countries, seeking to validate the results. In addition, comparative studies can be carried out once the database is updated with data during the pandemic and after it. Comparative studies can also be carried out between Latin American countries to find the similarities and differences with respect to consumption of food away from home.

## 6. Conclusions

In Peru, most families that consume food outside the home are nuclear, with the highest average spending for families defined as composite, followed by extended families. There is a higher CFAH when the head of household is a man. The CFAH spending that represents the income expressed in percentages is higher in one-person households. Non-poor households spend 7.20% of their income in CFAH, while extreme poor households spend 4.14%. Observing the expenditure on food consumed outside the home through the composition of households can allow a better approach to offer educational measures, and this information can be helpful to developers of educational issues. This study has made it possible to define that the households that spend the most on food consumed outside the home are compound households. They spend on average PEN 5520.67 per year, and PEN 460.06 per month. These families may have children and older adults in their composition, who need a lot of care in their diet. It is necessary to offer educational measures for these groups, which allow them to make better purchase decisions. More than 60% of the population incurred expenses from the consumption of food outside the home in 2019. More socioeconomic studies are required, and there is a need to establish how the expense considers the type of household and socioeconomic level. This period of growth and economic boom can be compared to recent or post-COVID-19 periods where there are restrictions and food prices are high. More research is warranted to determine sociodemographic characteristics of families to better understand their spending decision and take educational measures with these families.

## Figures and Tables

**Figure 1 foods-11-02547-f001:**
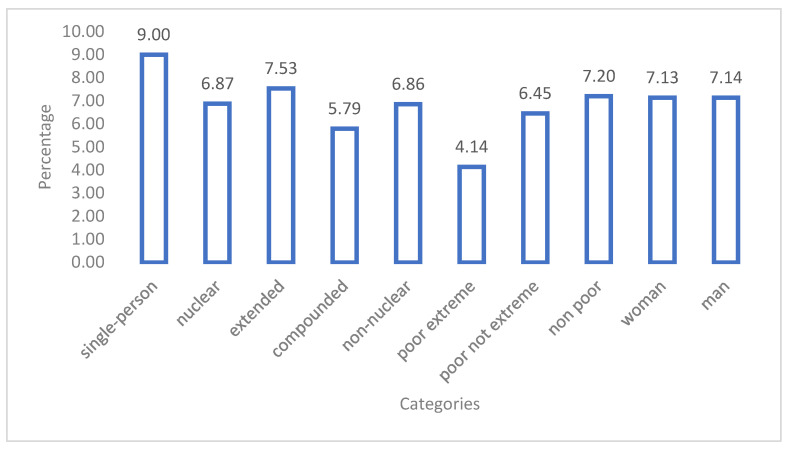
Food expenditure outside the household as percentage with respect to total net income (in PEN).

**Table 1 foods-11-02547-t001:** Participation of the head of household in food consumption away from home and household expenditure and per capita expenditure, ENAHO 2019.

Variable	Participation in Food Consumption outside the Home		Expenditure per Household (Annual)		Per Capita Spending (Annual)	
	Proportion	Standard Error	PEN (USD)	SD	PEN (USD)	SD
Head of household (gender)						
Male	0.813	0.002	3089.53 (925.01)	24.75	1060.43 (317.49)	23.26
Female	0.766	0.004	2938.92 (879.91)	38.05	1072.17 (321.01)	15.21
Difference	0.047 *		150.61 (45.09)		−11.74 (−3.51)	
	*p* = 0.0000		*p* = 0.0011		*p* = 0.3022	
Predominance						
Male	0.845	0.003	3744.68 (1121.16)	57.8	1305.18 (390.77)	25.63
Female	0.776	0.002	3175.06 (950.62)	41.04	928.75 (278.75)	13.69
Difference	0.69					
	*p* = 0.0000		*p* = 0.0000		*p* = 0.0000	
Presence of men						
Without men	0.637	0.007	1909.32 (571.65)	67.99	1076.67 (322.36)	46.61
With men	0.825	0.002	3561.57 (1066.34)	37.17	1062.33 (318.06)	13.57
Difference	0.188					
	*p* = 0.0000		*p* = 0.0000		*p* = 0.1482	
Presence of women						
Without women	0.779	0.007	3226.13 (965.91)	67.64	2860.05 (856.30)	84.87
With women	0.801	0.002	3030.06 (907.20)	21.81	907.61 (271.74)	10.77
Difference	0.0215					
	*p* = 0.0045		*p* = 0.0079		*p* = 0.0000	

Source: ENAHO 2019; (*) Statistically significant at *p*-value < 0.05. PEN = Peruvian sol, USD = US Dollar.

**Table 2 foods-11-02547-t002:** Types of households and the probability of food consumption outside the home and, household expenditure and per capita expenditure, ENAHO 2019.

Type of Household	Estimated Probability of Consumption	Age (Average) of the Head of Household (Years)	Average Annual Household Spending in CFAH [PEN (USD)]	Average Annual per Capita Household Spending on CFAH [PEN (USD)]
Single person	−0.363	57.38	2453.02 (734.44)	2453.02 (734.44)
Nuclear	0.405	49.54	3113.67 (932.24)	917.67 (274.75)
Extended	0.578	56.91	4290.40 (804.49)	804.49 (240.87)
Composite	0.29	52.42	4565.33 (818.03)	818.03 (244.92)
Non-nuclear	−0.006	50.64	3544.98 (1324.36)	1324.36 (396.51)
Total		52.19	3379.58 (1063.91)	1063.91 (318.54)

Source: ENAHO 2019. PEN = Peruvian sol, USD = US Dollar.

**Table 3 foods-11-02547-t003:** Coefficients and Marginal Effects of the Logit Model.

Variables	Coefficients	SD	Marginal Effects of the Logit dy/dx Model	SD
Gender				
Man	−0.129	±0.077	−0.035	±0.007
Age of household members *	−0.021	±0.002	−0.002	±0.000
predominance * ł	0.196	±0.057	0.0017	±0.006
extreme_major ł	−0.437	±0.067	−0.055	±0.007
extreme_minor ł	−0.459	±0.065	−0.060	±0.007
Type of household				
single-person–separated	−0.084	±0.126	0.013	±0.017
Single-person–divorced	−0.118	±0.282	0.083	±0.037
Single-person–widowed	−0.241	±0.126	−0.033	±0.018
Nuclear without children–cohabitant ł	−0.317	±0.164	−0.042	±0.025
Nuclear without children–married * ł	−0.534	±0.151	−0.050	±0.022
Nuclear with children–single ł	0.319	±0.281	0.101	±0.028
Nuclear with children–cohabitant * ł	0.562	±0.154	0.121	±0.021
Nuclear with children–married * ł	0.653	±0.152	0.154	±0.020
Nuclear with children–widowed * ł	0.489	±0.158	0.101	±0.019
Nuclear with children–divorced ł	0.465	±0.409	0.166	±0.037
Nuclear with children–separated * ł	0.363	±0.138	0.113	±0.019
Extended * ł	1.317	±0.153	0.202	±0.019
Compounded * ł	1.936	±0.211	0.174	±0.023
Non-nuclear	0.109	0.140	0.065	±0.019
Poverty				
poor not extreme * ł	0.983	0.102	0.182	0.017
not poor * ł	2.133	0.100	0.385	0.016
Constant	−0.017	0.196	0.98	0.136
pseudo R^2^	0.136		0.135	
Observations	34565		34565	

Source: ENAHO 2019; (*): significant coefficients *p* < 0.05 (ł): logit effect *p* < 0.05.

## Data Availability

The data presented in this study are available on request from the corresponding author.

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
