# Peer review of "Poverty, Household Structure and Consumption of Foods Away from Home in Peru in 2019: A Cross-Sectional Study"

_foods, 2022, doi:10.3390/foods11172547_

Round 1

Reviewer 1 Report

The paper is well organized. Few improvements could be added. More descritption about the methodology could be added. A discussion session about the generalization of obtained results could increase the validation process.  

 1. Please, think about correcting the title, eg Poverty, household structure and consumption of foods away  from home in Peru

2. Chapter 1 - I don't really understand the goal of the paper. What do you get when estimating CFAH expenditures by household type and the probability of CFAH according to the Logit model? I think it would be useful to explain it in more detail (in Chapter 1 or Chapter 2).

Please, define the structure of paper (at the end of chapter 1)

3. Chapter 2 - In addition to the cost of food consumed outside the home, does it matter how often? Whether yes or no, explain that in 1-2 sentences.

It might be useful to show the most important steps of the methodology as a figure

4. Table 2 - more clearly express the value – maybe, value in USD put in parentheses

5. I don't think you need to give an example from the US (lines from 222 to 226)? Please, think about it.

6. I think the conclusion needs to be strengthened. Why are the obtained results important and what do they mean?

7. Please, check the list of references - literature numbered 55 and 56 are mixed - correct where necessary.

Reviewer 2 Report

The paper is very interesting and overall well shaped. However, there are some aspects I would suggest considering in different parts, particularly the introduction, discussion, and conclusion.

A topic to include in the introduction can be cited to reinforce the purpose of the study. Please, check whether the outside-of-home consumption is treated in the EAT-Lancet commission (Willett et al., 2019 https://www.thelancet.com/journals/lancet/article/PIIS0140-6736(18)31788-4/fulltext). 

Other important aspects to include refer to

- considering the possible exploitation of charity services as outside-of-home consumption by poorer families - considering the possible difference between canteens and commercial restaurants that qualifies the type of choice - adjusting the conclusions accordingly and suggesting the utilization of the results in the food policy formulation.  

Moreover, for the methods section, I would suggest

- indicating the data source concern a randomly selected sample;

- explaining a little bit the difference between extended, composite, and non-nuclear households.

Reviewer 3 Report

Comment: Thank you for the opportunity to review the manuscript entitled “Poverty, household structure and consumption of foods away from home in Peru according to the 2019 National Household Survey”. Overall, the authors must enhance the utility of their research by adding managerial and public authorities’ implications. At present, few results are provided, with few discussions. Further, the “Literature review” section is missing. Further, the research relies on secondary data collected in 2019, prior to the COVID-19 pandemic: it means that such data should be explored and investigated in a more in-depth manner, considering that nowadays are somehow not updated. I have several major suggestions for the authors, which must be addressed in order to increase the scientific soundness of the research, as well as its originality and utility. 

Abstract: The abstract is almost clear, but some minor revisions should be addressed. The authors should revise the acronyms, since CFAH is repeated twice, whereas ENAHO is not useful (in the abstract). The authors should better clarify the purpose of the research, since at present it is not clear. 

Introduction: Introduction must be enhanced. 

First, lines 42-47 are not clear and, what is more, the authors list an incredible number of references without giving any comment. For instance: line 45, the authors cite references from 6 to 20, which means 15 references, without any particular comment.  

 Please, follow the Instruction for authors provided by MDPI. 

Some spelling and grammar mistakes must be corrected, as instance line 61: “… another important aspect I the trust…”. 

Considering that the section “Literature review” is missing, the authors must add more insights (and literature outcomes) related to households food consumption behavior, as well as consumers’ behavior with reference to food consumption away from home. Further, the authors must better clarify the purpose of the research, adding its novelty and its originality. To whom is the research addressed? Why is the research important? Besides, add some brief insights related to the methods applies. Please, refer to the subsequent articles, which investigate several aspects of consumers behavior out-of-home (e.g., expenditure, purchase, quality, frequency, food waste, etc.) and apply also logit models to investigate consumers’ behavior. 

Amicarelli, V., Lagioia, G., Sampietro, S. and Bux, C. (2022). Has the COVID-19 pandemic changed food waste perception and behavior? Evidence from Italian consumers. Socio-Economic Planning Sciences, 82, Part A, 101095. 10.1016/j.seps.2021.101095

Law, C., Smith, R., Cornelsen, L. (2022). Place matters: Out-of-home demand for food and beverages in Great Britain. Food Policy, 107, 102215. https://doi.org/10.1016/j.foodpol.2021.102215

Vargas-Lopez, A., Cicatiello, C., Principato, L., Secondi, L. (2022). Consumer expenditure, elasticity and value of food waste: A Quadratic Almost Ideal Demand System for evaluating changes in Mexico during COVID-19. Socio-Economic Planning Sciences, 82, Part A, 101065,https://doi.org/10.1016/j.seps.2021.101065.

Alaimo L.S., Fiore M., & Galati A. (2020). How the COVID-19 pandemic is changing online food shopping human behaviour in Italy. Sustainability, 12(22), 1-18. 10.3390/su12229594.

No mention is done to the COVID-19 pandemic. It represents a criticality of the analysis. Please, consider adding some additional insights related to the COVID-19, at least by citing such a limitation of the research and possible future research directions. 

Methodology: The section “Methodology” must be enhanced. No references have been provided in the section. Please, justify the choice of the variables, as well as the construction of the research and the methods adopted to analyze data by referring to previous (and successful) studies, as well as statistical handbooks and other official documents. 

For examples, in lines 101-110, is it possible to refer to previous studies that have distinguished between nuclear homes, extended households, composite households, etc.?

Line 111: “…to construct the indicator of the present study…”? Which is the indicator you refer to? Please clarify, since no mention is done to any “indicator” prior to line 111. 

Line 112. How is the “expenditure” calculated on the basis of the ENAHO survey? 

It is still not clear where the data come from. Perhaps, from ENAHO (line 89)? If this is the right answer, please add several more details related to the data quality, the characteristics of the survey, as well as all the variables investigated in such a survey. The authors declare: “This variable allowed us to know if a household, with its various characteristics, has consumer food outside the home during 2019” (lines 90-91). Which is the structure of the survey? How is the survey distributed? How is the survey representative of the Peruvian population? etc. Please, add more details. 

Line 124. “This database contains a series of variables that are related to the objective of this study”. Which variables? At present, the description of the data quality is insufficient and superficial. 

Still, the aim of the research is not clear, so please try to do your best, from the very beginning, to highlight the purpose of the research. 

Construction of indicators. Could the authors provide a table which summarizes the indicators and the considered variables involved? 

Results and Discussion. “Results” are clear, but “Discussion” are quite poor. I cannot see any reference to theoretical or managerial implications. Besides, are there any utilities for public authorities or policy makers? The authors declare that “the results of this study allow researchers and authorities to go beyond the economic data and create better public policies”. Please, provide some solutions to address public policies. Further, the authors should discuss limitations of the present research and future research directions. In addition, I expect reading some comparisons of the results of the present research with other international realities. 

General comment: Several references are cited in the reference list, but not in the main body of the text. Please, revise. 

Round 2

Reviewer 3 Report

Comment: Thank you for the opportunity to review the revised version of the manuscript entitled “Poverty, household structure and consumption of foods away from home in Peru in 2019: a cross-sectional study”. Although the authors have revised the manuscript, I still have some major concerns related to the research. The section “Introduction” must be substantially revised, since at present it is not informative and contains a huge number of “uncommented” studies, which make the article rather inconsistent. See the comments below and revise the manuscript in a coherent manner. 

First, the abstract accounts for more than 350 words, whereas its maximum should be approx. 200 words. 

I cannot understand lines 55-62. The authors are supposed to introduce food choices, food consumption behaviors, whereas I cannot understand why the authors describe (in detail) the reactions between free radicals and other macromolecules towards the formation of reactive oxygen species, which only appear in such lines are not useful for the research. Moreover, the authors cite at line 62 an incredible number of references (i.e., 2,8,10-36). This requires a major revision, since I cannot understand the utility of citing so many references to justify a useless topic. Moreover, as already stated in the first revision, the authors cite several references without providing any comment to such references. For instance, line 63, where [3,37-45] are cited without any comment. I have a possible solution for the authors: reduce the number of these references and discuss them. On the other way, I cannot see the utility of citing such a number of studies without providing any additional detail. 

Section “Literature review” seems useful and in line with the aims and scope of the research. 

Lines 147-149 are not grammatically clear. 

Lines 187-194. Please, try to make clearer the list of “types of household”, for instance by introducing each “type” with a letter or a number. 

Figure 2 do not represent a “graphical research framework”. Please, try to provide a scientific graphical representation of the research framework, which includes all the steps carried out in the research. At present, Figure 2 is neither useful nor informative. 

“Discussion” have not been substantially revised by the authors. Still, the section is quite poor. 
